# Antifouling Systems Based on a Polyhedral Oligomeric Silsesquioxane-Based Hexyl Imidazolium Salt Adsorbed on Copper Nanoparticles Supported on Titania

**DOI:** 10.3390/nano13071291

**Published:** 2023-04-06

**Authors:** Alessandro Presentato, Eleonora La Greca, Luca Consentino, Rosa Alduina, Leonarda Francesca Liotta, Michelangelo Gruttadauria

**Affiliations:** 1Dipartimento di Scienze e Tecnologie Biologiche, Chimiche e Farmaceutiche, Viale Delle Scienze, Edificio 17, I-90128 Palermo, Italy; alessandro.presentato@unipa.it (A.P.); luca.consentino@ismn.cnr.it (L.C.); valeria.alduina@unipa.it (R.A.); 2Istituto per lo Studio dei Materiali Nanostrutturati (ISMN)-CNR, Via Ugo La Malfa 153, I-90146 Palermo, Italy; eleonora.lagreca@ismn.cnr.it

**Keywords:** marine biofouling, polyhedral oligomeric silsesquioxane, imidazolium salt, copper nanoparticles, titania

## Abstract

The reaction of octakis(3-chloropropyl)octasilsesquioxane with four equivalents of 1-hexylimidazole or 1-decylimidazole gave two products labelled as HQ-POSS (hexyl-imidazolium quaternized POSS) and DQ-POSS (decyl-imidazolium quaternized POSS) as regioisomer mixtures. An investigation of the biological activity of these two compounds revealed the higher antimicrobial performances of HQ-POSS against Gram-positive and Gram-negative microorganisms, proving its broad-spectrum activity. Due to its very viscous nature, HQ-POSS was adsorbed in variable amounts on the surface of biologically active oxides to gain advantages regarding the expendability of such formulations from an applicative perspective. Titania and 5 wt% Cu on titania were used as supports. The materials 10HQ-POSS/Ti and 15HQ-POSS/5CuTi strongly inhibited the ability of *Pseudomonas* PS27 cells—a bacterial strain described for its ability to handle very toxic organic solvents and perfluorinated compounds—to grow as planktonic cells. Moreover, the best formulations (i.e., 10HQ-POSS/Ti and 15HQ-POSS/5CuTi) could prevent *Pseudomonas* PS27 biofilm formation at a certain concentration (250 μg mL^−1^) which greatly impaired bacterial planktonic growth. Specifically, 15HQ-POSS/5CuTi completely impaired cell adhesion, thus successfully prejudicing biofilm formation and proving its suitability as a potential antifouling agent. Considering that most studies deal with quaternary ammonium salts (QASs) with long alkyl chains (>10 carbon atoms), the results reported here on hexylimidazolium-based POSS further deepen the knowledge of QAS formulations which can be used as antifouling compounds.

## 1. Introduction

The attachment and thrift of microorganisms on boat surfaces immersed in seawater cause the growth of plants and animals, generating abnormal weight increases and high frictional resistance that, in turn, result in higher fuel consumption and much lower overall sustainability [1,2]. The higher cost associated with these events also relates to the higher frequency of dry dock work. Furthermore, the association of different marine species with ship hulls, especially for vessels sailing worldwide, can introduce invasive species into a specific marine environment. Maintaining fouling-free surfaces is challenging since thousands of marine organisms are responsible for biofouling and many of these can adapt to environmental changes. Innovative and sustainable solutions to this topic are still of paramount interest for commercial reasons and the environment’s safety. Thus, developing new cost-effective and eco-friendly formulations featuring powerful antifouling properties is highly desirable. A class of compounds extensively investigated for such a purpose is quaternary ammonium salts (QASs). Molecules and polymers based on QASs exert a killing effect against a wide range of microorganisms, including Gram-positive and Gram-negative bacteria, with the nature of their cation (aliphatic, heteroaromatic) and the length of their alkyl chains being responsible for modulating their antibacterial activity [3,4,5,6,7,8,9,10]. In addition to QASs, polyhedral oligomeric silsesquioxane (POSS)-based molecules are gaining momentum in this context. Quaternization of dimethyl-n-octylamine by octakis(3-chloropropyl)octasilsesquioxane gave the corresponding salts in which almost all eight chloropropyl groups of the silsesquioxane cage, or a part of these groups, were transformed into ionic quaternary ammonium chloride functions. The product containing ca. 50% ionic groups exhibited a broad-spectrum of bactericidal activity [11]. Alkyl QAS-functionalized POSS compounds with different alkyl chain lengths, counterions, and cross-linked polydimethylsiloxane networks containing dispersed QAS-POSS molecules displayed antimicrobial activity [12]. Sum frequency generation vibrational spectroscopy (SFG) was used to investigate the molecular surface structures of two types of QAS-incorporated polydimethylsiloxane (PDMS) systems in different chemical environments, giving more insights into their antifouling properties [13]. Several ammonium POSS compounds varying in their extent of quaternization and alkyl chain length showed antimicrobial activity, which was strongly dependent on their quaternized POSS composition. Quaternized POSSs characterizing a relatively low degree of quaternization and long alkyl chain lengths provide the highest antimicrobial activity. Contrarily, the low antimicrobial activity of POSSs having a relatively high level of quaternization was ascribed to the aggregation of such molecules in solution, inhibiting their diffusion into bacterial cell surface structures and affecting the mobility of their alkyl chains in aqueous environments [13,14]. POSSs modified with 1,2,3-triazolium functional groups showed bactericidal activity [15]. A POSS cage can be used as a polymer core thanks to its eight functional groups in three dimensions and employed in the synthesis of sulfobetaine/quaternary ammonium-modified star-shaped poly[2-(dimethylamino)ethyl methacrylate]-based copolymers [16]. A POSS structure was also applied to obtain hydrogels containing octa-betaine esters [17]. Moreover, a POSS nanocage used as a bridge in a multifunctional antimicrobial agent containing light-harvesting oligo(*p*-phenylenevinylene) electrolytes (OPVEs) multi arms and a central porphyrin core was designed [18].

In addition to organic molecules, metal nanoparticles play a role in this field. The antifouling and antimicrobial activity of Cu nanoparticles supported on silica and titania are well-known [19]. The tunable surface properties of silica and titania (e.g., high specific surface area, hydrophobic/hydrophilic balance) and their high thermal, chemical, and biological stability make such oxides suitable as metal supports. Silica, titania, and mixed silica-titania powders were used as supports for loading copper, silver, or copper-silver (5 wt%), and these materials displayed antimicrobial properties [20].

Here, two new quaternary ammonium salts based on a POSS cage structure and their preliminary biological evaluation as antimicrobial agents are presented. Then, the most active one was used in conjunction with titania or titania copper-based powders by simple adsorption to obtain an easily manageable powder possessing antimicrobial properties. Biological investigations on these materials are also reported.

## 2. Materials and Methods

### 2.1. Sample Preparation

#### 2.1.1. Synthesis of HQ-POSS and DQ-POSS

Octakis(3-chloropropyl)octasilsesquioxane 1 (300 mg, 0.289 mmol) was placed in a vial and dissolved in toluene (150 μL), along with adding 1-hexylimidazole (185 mg, 4 equiv.) or 1-decylimidazole (241 mg, 4 equiv.). The solution was heated at 90 °C for 19 h and formed, after this time, a very viscous phase. The latter was dissolved in a minimum amount of chloroform by gentle heating and sonication and transferred to a round bottom flask. This procedure was repeated three times and the solvent was removed under reduced pressure to give very viscous compounds (nearly quantitative yield, a small amount of solvent remained).

#### 2.1.2. Synthesis of 5 wt% Cu/TiO_2_ and HQ-POSS-Loaded Cu5 wt%/TiO_2_ Samples

The 5 wt% Cu/TiO_2_ material was prepared by the wetness impregnation method of commercial TiO_2_ oxide (Sigma-Aldrich) with Cu(NO_3_)_2_·2.5H_2_O dissolved in the minimum volume of distilled water. The impregnation of the solid was performed by adding the copper solution to the powder drop-by-drop, by getting it wet but not soaked. The procedure was repeated until all the precursor was added. Then, after drying at 120 °C overnight, the resulting material was labelled as 5CuTi and calcined at 500 °C for 2 h with a heating ramp of 5 °C/min.

Based on the obtained support, a series of HQ-POSS/CuTi samples with loadings ranging from 2 to 30% by weight were prepared, as listed in Table 1. All samples were prepared by impregnating the support with chloroform solutions containing the right amount of the HQ-POSS molecule. This molecule was added to a flask with CHCl_3_ solution and stirred (about 600 rpm) until complete dissolution. Subsequently, the amount corresponding to the percentage by weight of 5%Cu/TiO_2_ was added, and it was left to stir to promote its complete impregnation. After this time, the solution was dried by rotavapor at 40°C. The dry residue thus obtained was recovered and signed (Table 1). As a reference, a sample with a selected loading, 10 wt% HQ-POSS, was prepared on commercial TiO_2_ previously calcined at 500 °C for 2 h (5 °C/min). The resulting materials were labeled as xHQ-POSS/5CuTi (x = 2, 10, 15, 30 wt%) and 15HQ-POSS/Ti.

### 2.2. Sample Characterization

Elemental analysis of the 5CuTi sample was carried out by atomic emission spectroscopy (MP-AES 4200; Agilent Technologies Italia S.p.A., Cernusco sul Naviglio (MI), Italy) after treatment in acidic solution with H_2_SO_4_ and HNO_3_ at 250 °C. The real chemical composition corresponded well to the nominal one with a 5.0 wt% ± 5%.

The specific surface area (SSA) and pore volumes of selected samples were determined from N_2_ adsorption–desorption isotherms at −196 °C, using Micromeritics ASAP 2020 equipment through the Brunauer–Emmett–Teller (BET) method in the standard pressure range 0.05–0.3 P/P_0_. Before the measurements, the samples were degassed at 250 °C for 2 h. By analysis of the desorption curves, using the Barrett–Joyner–Halenda (BJH) calculation method, pore size distributions were also obtained [21]. The precision of the SSA values was ±1%, ±0.01 for pore volume and ±0.1 for the mean pore size values. 

X-ray diffraction (XRD) patterns were recorded with a D5000 diffractometer (Bruker AXS; Bruker, Karlsruhe, Germany), employing Ni-filtered Cu Kα radiation in the 2θ range between 20° and 80° with a 0.05° step size and time per step of 20 s. The assignment of the crystalline phases was based on the ICSD files. 

The materials’ thermogravimetric analyses (TGAs) were performed in a TGA/DSC1 STAR System from Mettler Toledo Inc.( Mettler-Toledo S.p.A., Milano, Italy) For each analysis, the sample (15 mg) was subjected to a pretreatment in airflow (30 mL/min) from 25 °C to 100 °C with a heating rate of 10 °C/min and a holding time at 100 °C for 30 min, to remove any eventual residual solvent. Then, the temperature was increased from 100 to 1000 °C under airflow (30 mL/min), and the weight loss occurring during this step was considered for calculating the organic weight content of the HQ-POSS-based Cu_5_TiO_2_ materials. The precision of the mass loss values was ±0.001 and ±2% of the wt% of HQ-POSS in the samples.

#### 2.2.1. Antimicrobial Activity of Quaternized POSS Compounds

The minimal inhibitory concentration (MIC), i.e., the lowest amount of an antimicrobial compound that visually impairs bacterial growth, of HQ- and DQ-POSS against the Gram-positive (i.e., *Bacillus cereus*) and Gram-negative (i.e., *Stenotrophomonas maltophilia*) marine environmental isolates was tested as described elsewhere [22]. Briefly, pre-cultures of selected microorganisms were carried out by inoculating a single bacterial colony in the Luria Bertani ((LB); composed (g L^−1^) of sodium chloride (NaCl; 10), tryptone (10), and yeast extract (5)) medium and grown for ca. 16 h at 30 °C under shacking conditions (180 rpm). Then, 5 × 10^5^ colony-forming units per milliliter of culture (CFU mL^−1^) were inoculated in fresh LB medium with increasing concentrations (up to 500 μg mL^−1^) of HQ- or DQ-POSS. Bacterial cultures were incubated for 24 h as previously described. Afterward, the bacterial cultures’ optical densities at 600 nm (OD_600nm_) were evaluated using a UV–Vis spectrophotometer (SPECTROstar^®^ Nano; BMG Labtech, Milan, Italy).

Kill curves were carried out to evaluate the minimal bactericidal concentration (MBC), i.e., the lowest concentration of a bactericidal agent able to kill off all microbial cells of a given bacterial species, of HQ and DQ-POSS. Bacterial strains were pre-cultivated and inoculated as previously described in the presence of 1, 2.5, 5, or 7.5 μg mL^−1^ of quaternized POSS. After a 24 h challenge, bacterial cultures were serially diluted and each dilution was spotted onto LB-agar (15 g L^−1^) recovery plates, which were incubated at 30 °C under static conditions to allow bacterial colony growth [23]. The average value (n = 3) of the number of CFUs that survived the 24 h challenge is expressed in a logarithmic scale as a function of the concentration of quaternized POSS. In the case of adsorbed HQ-POSS onto copper/titania (CuTi) or titania (Ti) supports, *Pseudomonas* PS27 was used as a Gram-negative strain instead of *S. maltophilia*.

The most promising adsorbed HQ-POSS (i.e., 10HQ-POSS/Ti and 15HQ-POSS/5CuTi) were tested to unveil their ability to prevent *Pseudomonas* PS27 biofilm formation, as described elsewhere [24]. *Pseudomonas* cells were pre-cultivated as described earlier and inoculated at an initial biomass load corresponding to 1 × 10^7^ CFU mL^−1^. Afterward, the bacterial culture was aliquoted (200 μL) in a 96-well microtiter plate and amended with 250 μg mL^−1^ (corresponding to the best biocidal concentration against *Pseudomonas* PS27 cells) of either 10HQ-POSS/Ti or 15HQ-POSS/5CuTi. Thus, bacterial cultures were incubated at 37 °C for 24 h under static conditions to allow bacterial cells to grow as a biofilm. Then, planktonic cells were disposed of and each well was washed 3 times with a sterile saline (NaCl 0.9% *w*/*v*) solution to loosely remove adherent cells. Thus, *Pseudomonas* biofilms were stained (for 15 min at room temperature) using 200 μL of an aqueous solution of crystal violet (0.2% *w*/*v*). After staining, excesses of unreacted crystal violet were removed through 5 washing steps with 200 μL of saline solution, with the plate left to dry for 20 min at 65 °C. Crystal violet bound to *Pseudomonas* cells was solubilized by adding 200 μL of an aqueous solution of acetic acid (30% *v*/*v*) and the absorbance of each well was read at 595 nm (Synergy HT; Biotek). Unchallenged *Pseudomonas* biofilm was used as a reference (positive control) and wells containing uninoculated medium, as a negative control. Thus, the percentage of biofilm formation was calculated as follows:(1)$$Biofilm\;formation\;\left(\%\right)=\frac{absorbance\;positive\;control-absorbance\;negative\;control}{absorbance\;positive\;control}\times 100$$

All the experiments were performed at least in triplicate and data are reported as the average value (n = 3) with standard deviations.

All reagents were purchased from Merck Life Science S.r.l. (Milan, Italy). 

#### 2.2.2. HPLC/MS Analysis

HPLC/MS analysis was performed by adapting previously reported methods [25,26]. Samples for HPLC (Agilent 1260 Infinity) were prepared by solubilizing the compound HQ-POSS (0.4 mg) in MeOH (1 mL). Water and acetonitrile were of an HPLC/MS grade. Formic acid was of analytical quality. A reversed-phase Phenomenex Luna Omega 5 µm Polar C18 100 column (150 mm × 2.2 mm) with a Phenomenex C18 security guard column (4 mm × 3 mm) was used. The injection volume was 10 µL. The eluate was monitored through mass total ion count (MS TIC) and UV (250 nm). Mass spectra were obtained on an Agilent 6540 UHD accurate-mass quadrupole time-of-flight (Q-TOF) spectrometer equipped with a dual AJS electrospray ionization (ESI) source working in positive mode. Nitrogen N_2_ was employed as desolvation gas at 300 °C and a flow rate of 8 L/min. The nebulizer was set to 45 psig. The sheath gas temperature was set at 350 °C and a flow of 12 L/min. A potential of 3.2 kV was used on the capillary for the positive ion mode. The fragmentor was set to 175 V. MS spectra were recorded in the 150–3200 m/z range. Mass spectrum data were analyzed by using the software MassHunter Qualitative Analysis B.06.00 (Agilent Technologies Italia S.p.A. Cernusco sul Naviglio (MI), Italy).

## 3. Results and Discussion

### 3.1. Synthesis and Characterization of HQ-POSS and DQ-POSS 

Based on the antimicrobial activity of 1,3-disubstituted imidazolium salts and considering the usefulness of the POSS cage and the biological activity of quaternized POSS molecules, the possibility of easily synthesizing new alkylimidazolium-POSS salts to investigate their antimicrobial activity was envisioned. Then, due to the high polar nature of such imidazolium-POSS salts and to obtain powders easily manageable and mixable with the proper paint, the obtained salts were adsorbed on the surface of materials suitable for this purpose. 

The synthesis started with octakis(3-chloropropyl)octasilsesquioxane **1** by its reaction with four equivalents of 1-hexylimidazole or 1-decylimidazole (Figure 1). The use of four equivalents depended on the fact that a high level of quaternization can give substrates with low antimicrobial activity. Quaternization reactions were carried out in concentrated toluene solutions by simply heating a solution of POSS **1** and alkylimidazole, resulting in very viscous products. The two products were labelled as HQ-POSS (hexyl-imidazolium quaternized POSS) and DQ-POSS (decyl-imidazolium quaternized POSS). Tetra-substitution can give a mixture of inseparable regioisomers that were used as obtained.

The ^1^H NMR spectrum of HQ-POSS was in accordance with the proposed structure (Figure 1). Protons of the imidazolium ring are visible at 10.51 ppm and in the range of 7.30–8.60 ppm. Methylene groups (H-6 and H-7) linked at the nitrogen atoms resonate at 4.26 and 4.48 ppm. Protons H-1′ (CH_2_Cl) resonate at 3.52 ppm and protons H-2′ could be assigned to the shoulder at ca. 1.99 ppm, whereas protons H2′ and H-5 could relate to the signal at 1.86 ppm. The signal at 1.22 ppm could be assigned to H-2, H-3, and H-4 protons, the signal at 0.85 ppm, to H-1, and the signal at 0.76, to methylene groups linked to silicon (H-9 and H-3′). Considering the signals of H-6 and H-7 protons and the signal of H-2′, it was possible to estimate a ca. 1:1 ratio between chloropropyl and 1-hexyl-3-propyl-1H-imidazol-3-ium chains that was in agreement with a tetra-substituted POSS. This NMR pattern included the presence of possible regioisomers and did not exclude the possible presence of mono-, di-, and tri-substituted POSSs. The ^1^H NMR spectrum of DQ-POSS (Figure 2) showed the same signal patterns. In the case of DQ-POSS, the ratio between chloropropyl and 1-decyl-3-propyl-1H-imidazol-3-ium chains was about 60:40, indicating that a tetra-substitution pattern was less complete. This result can be explained by the presence of the more hindered decyl chain.

HPLC/MS analysis of HQ-POSS showed, in addition to the presence of tetra-substituted HQ-POSS isomers, the presence of at least two tri-substituted HQ-POSS, the expected three di-substituted HQ-POSS [27], and the mono-substituted HQ-POSS (see Appendix A).

### 3.2. Biological Evaluation of HQ-POSS and DQ-POSS

The next step investigated the antimicrobial activity of HQ-POSS and DQ-POSS. The quaternized HQ- and DQ-POSS exhibited different antimicrobial activity against the Gram-negative and -positive marine bacteria *S. maltophilia* and *B. cereus* strains, respectively (Table 2). The antimicrobial activity of these quaternized POSSs appears to depend on the alkyl chain length of QAS groups. Specifically, HQ-POSS showed an enhanced antimicrobial potential (MIC < 10 μg mL^−1^) compared to DQ-POSS, which exhibited an impairment of bacterial growth at a concentration as high as 500 μg mL^−1^.

According to the MIC values of the HQ- and DQ-POSS formulations (Table 2), lower concentrations were tested to evaluate whether the POSS compounds’ MBC corresponded to their MIC. Direct estimation of bacterial survival (CFU ml^−1^) due to the challenge exerted by quaternized POSSs underlines the far greater proficiency of HQ-POSS in inhibiting bacterial growth (Figure 3). As a result, HQ-POSS exhibited great antimicrobial activity since the lowest concentration (1 μg mL^−1^) was sufficient to carry out a 100% killing effect—corresponding to the MBC value—toward Gram-negative bacterial cells within the 24 h challenge (Figure 3a). Gram-positive bacterial cells appeared instead to be more resilient against the biocidal action of HQ-POSS since it was necessary to increase its concentration up to 2.5 μg mL^−1^ to observe a complete killing effect. Indeed, the lowest HQ-POSS concentration (1 μg mL^−1^) only determined a slight decrease in CFU evaluated (Figure 3b). Instead, the formulation DQ-POSS did not affect bacterial survival (Figure 3). 

From a physicochemical standpoint, QASs are compounds featuring a single head, an aliphatic chain, and a counterion. All these structural elements contribute to determining their antimicrobial potential against bacterial species growing planktonically or forming a biofilm [28]. Research to date suggests that the mechanism of action of such compounds relies on severe perturbations of the cell membrane and alteration of their physiological fluidity, which lead to protein membrane instability, causing the malfunctioning of their transport and energy-transducing systems, sensing, signal transductions, migration, and adhesion among others [29,30]. Alkyl chain length is a structural element that strongly influences the antimicrobial potential of QASs. The optimal alkyl tail length is specific for each set of structures and is related to the other fragments of the molecule. Generally, it can be within C10-C14 or even longer (C18), but it can vary depending on the number of charges: C12 and longer for mono-QASs and C10-C12 for bis-QASs. Nevertheless, in selected cases, those with tails of C10 and shorter demonstrated the highest activity [31]. The different antimicrobial behavior featuring short- and medium to long-chain QASs relies on the ability of the latter to better intercalate within the bacterial cell membrane [28,32]. A separate dependency for short- (n < 10) and long- (n >10) chain substituents in the antimicrobial potential of alkyltrimethylammonium bromides possessing alkyl chain lengths ranging from 5 to 22 carbon atoms against *Staphylococcus aureus*, *Pseudomonas aeruginosa*, and *Saccharomyces cerevisiae* strains has already been demonstrated [33]. This was interpreted in terms of distinct mechanisms of action, binding site, and/or physicochemical properties for extreme members of the series. The evidence reported here further enlarges the spectrum of QAS-based formulations (i.e., those featuring short-chain QASs), which can be used as bactericidal agents.

However, it is worth mentioning that when it comes to quaternized POSS, its antimicrobial activity is dependent on both its alkyl chain length and extent of quaternization, where a low degree of the latter alongside a long alkyl chain QAS confers good lipophilicity and diffusivity to the final QAS-POSS, which can effectively bind to the microbial cell surface deploying its antimicrobial potential [14]. An opposite behavior features QAS-POSS holding a high level of quaternization with long-chain QASs, which can interact with one another through hydrophobic interactions occurring between their alkyl chains in an aqueous solution, leading to aggregation phenomena of these molecules and the consequent formation of large-sized particles that diffuse slowly and are not as bioavailable as POSS with a low degree of quaternization featuring a short-chain QAS [12,14]. Thus, the latter has the advantage of diffusing, binding, intercalating, and permeating the outer layer structures of bacterial cells better compared to POSSs quaternized with long-chain QASs, enhancing cell death events, and reasonably explaining why DQ-POSS did not highlight any antimicrobial potential against the bacterial strains tested (Figure 3).

### 3.3. Preparation of Adsorbed HQ-POSS-Based Materials and Their Biological Evaluation

Considering the highly viscous nature of HQ-POSS and its very interesting biological results compared to DQ-POSS, a variable amount of HQ-POSS was adsorbed on the surface of biologically active oxides [20] to gain advantages regarding the expendability of such formulations from an applicative perspective (e.g., enhanced dispersibility into paints for boat surfaces). The specific surface areas (SSAs), pore size distributions, and pore volumes of the xHQ-POSS/5CuTi materials are listed in Table 3. The parent TiO_2_, TiO_2_ after calcination at 500 °C and the 10HQ-POSS/Ti samples are also enclosed. 

The data revealed that the textural properties of all samples doped with the HQ-POSS molecule showed significant differences compared to the calcined TiO_2_ and the parent material (5%CuTi). Firstly, from the comparison between commercial TiO_2_ oxide, as received, with the material after calcination, the latter resulted in a lowering of the SSA values along with pore volumes, while pore sizes increased. Such findings agree with the shrinkage of the external surface and porosity of titania as a consequence of the calcination process, which likely causes the removal of the structural water and surface OH groups, the collapse of the smallest pores, and a shift of the mean pore size distribution toward higher values. A similar trend arises by comparing the properties of 5CuTi with the parent TiO_2_. Indeed, the 5CuTi sample was characterized by an SSA of 52.8 m^2^/g vs. 82 m^2^/g for the titania, the mean pore size shifted toward higher values, and the pore volume decreased. The reported data suggest that after copper nitrate impregnation over titania and further calcination, the CuO nanoparticles formed may have partially covered the external surface of the support and blocked some pores. Accordingly, XRD patterns registered on the 5CuTi sample revealed crystalline peaks assigned to the CuO phase along with the typical features of titania, as anatase and rutile, which were confirmed by cross-referencing it with ICSD files (see Appendix A). 

By depositing increasing loadings of the complex HQ-POSS structures onto the 5CuTi, further decreases in specific surfaces, pore volumes, and pore sizes arose as a function of the loading. The textural properties of 30HQ-POSS/5CuTi were not determined due to the high loading of the organic moieties that destabilized the system under the vacuum pretreatment necessary to perform N_2_ physisorption.

The results of the thermogravimetric analysis (Figure 4), which investigated the thermal stability of the xHQ-POSS/5CuTi composites, confirm the effective modification of 5CuTi oxide by HQ-POSS and estimate effective HQ-POSS loading. This analysis reveals that the most weight loss (71%) occurred in the HQ-POSS molecule in the temperature range of 150 to 700 °C. The first weight loss peak below 200 °C can be attributed to the removal of water contained in the sample, while the second weight loss peak in the range of 250–700 °C corresponds to the decomposition of the organic part. As the HQ-POSS loading decreased, the percentage of mass lost also decreased, which is consistent with the percentage by mass of the molecule loaded on 5CuTi. The remaining residual mass was likely due to the formation of SiO_2_ oxide deposited over the parent oxides, 5CuTi or titania. 

To further prove the suitability of such formulations in inhibiting bacterial cell proliferation, the Gram-negative microbe tested, in this case, was *Pseudomonas* PS27, an environmental isolate far more resilient to environmental stress than *S. maltophilia*, utilized to screen the antimicrobial potential of HQ- and DQ-POSS. Indeed, this *Pseudomonas* strain can use organic solvents (i.e., octane and a mixture of ethanol-octane) as its only sources of carbon and energy, which concomitantly stimulate its biotic accumulation of the toxic xenobiotic compound perfluorohexane sulfonate [34].

As a result, 10HQ-POSS/5CuTi, 15HQ-POSS/5CuTi, and 10HQ-POSS/Ti are the formulations that most affected bacterial survival (Figure 5), although some differences between Gram-positive and -negative strains were observed. The former resulted in an enhanced susceptibility toward the most effective formulations since 50 μg mL^−1^ (15HQ-POSS/5CuTi) or 100 μg mL^−1^ of (10HQ-POSS/5CuTi or 10HQ-POSS/Ti)—corresponding to the MBC values—was sufficient to completely kill *B. cereus* cells within the 24 h challenge (Figure 5b). The 10QH-POSS/5CuTi and 10HQ-POSS/Ti formulations exerted the same effect as 15HQ-POSS/5CuTi at a higher concentration, likely due to less HQ-POSS having been adsorbed onto the supports. In the case of the Gram-negative strain, these formulations showed their antimicrobial potential at a concentration of 250 μg mL^−1^, highlighting a dramatic impairment of *Pseudomonas* PS27 cell growth, which resulted in a CFU reduction of ca. 6 (10HQ-POSS/5CuTi) or 8 (15HQ-POSS/5CuTi and 10HQ-POSS/Ti) logarithmic units (Figure 5a). The enormous drop in CFUs already observed at 100 μg mL^−1^ of 10HQ-POSS/5CuTi and 15HQ-POSS/5CuTi compared to 10HQ-POSS/Ti was likely dependent on the fastest-release kinetic of HQ-POSS from CuTi supports compared to those containing only titania, which would explain the reason why 10HQ-POSS/Ti only exerted its full antimicrobial potential at the highest concentration tested (250 μg mL^−1^). However, in the case of *Pseudomonas* PS27, MBC values for the most active formulations were not found. This aspect corroborates the good adaptability of this environmental isolate to organic xenobiotics that, similarly to QASs, can compromise bacterial cell membrane fluidity and functionality, leading to cell death [35,36]. These results are of some importance, considering (i) the resilience of this environmental isolate against toxic organic xenobiotics and (ii) the far greater resistance of Gram-negative strains to antimicrobial compounds. Indeed, these bacterial strains possess an outer cell membrane that acts as an additional physical barrier that opposes incoming stressors [37,38]. The 2HQ-POSS/5CuTi formulation and the supports alone (i.e., 5CuTi and Ti) did not exhibit any influence on the bacterial cell growth of both strains (Figure 5), likely due to the low weight percentage of HQ-POSS adsorbed onto the support, or because of the short incubation time of the assay to allow supports to elicit their antimicrobial activity. In line with this hypothesis, Calabrese and colleagues [20] successfully demonstrated the bacteriostatic effect of titania supports doped with copper nanoparticles against three marine bacterial isolates over nine days of bacterial incubation.

The vast majority of bacterial species feature the ability to grow, forming complex structural communities known as biofilms, through a highly regulated process involving intercellular signaling, which overall regulates bacterial growth and cell behavior [39]. After bacterial cells adhere to a surface and aggregate with each other, the microbes grow as a biofilm and encapsulate themselves into an extracellular polymeric substance. In this context, bacterial cells acquire intriguing biological traits that correlate with stratified cell metabolism, genetic, and physiological features and increased antimicrobial tolerance, resistance, and virulence. All these factors contribute to improving bacterial cell fitness and resilience to external stress [40,41,42]. Moreover, biofilms represent a major cause of biofouling in aquaculture, targeting culture species and infrastructures with significant economic losses [43,44]. Thus, it is crucial to combat biofilm development at the very beginning of its formation, for instance, by using antifouling compounds. Moreover, *Pseudomonas* species are of extreme concern for such a matter since they are proficient biofilm producers, opportunistic pathogens for immunocompromised humans, as well as the cause of spoilage in aquatic environments [43,45,46]. Since POSS materials do not feature antimicrobial action *per se*, their functionalization with QASs can lead to efficient antifouling agents. The killing effect would rely on the cationic nature (i.e., the positively charged nitrogen atom) of QAS and the amphiphilic character of its chain. These chemistry features allow QASs to interact, with a high affinity, with the bacterial cell membrane [47,48]. Thus, quaternary ammonium groups first adsorb to the bacterial cell surface through electrostatic interactions, which is emphasized in the case of POSSs functionalized with QASs due to the small size of the former and, consequently, an increased charge density similar to dendrimers [49]. Then, these compounds diffuse through superficial bacterial cell structures. Finally, QASs’ alkyl chains intercalate the cytoplasmic membrane, altering its physiological integrity, and leading to the leakage and shedding of intracellular and superficial macromolecules, thus compromising cell survival [50]. Here, the best antimicrobial formulations (i.e., 10HQ-POSS/Ti and 15HQ-POSS/5CuTi) strongly inhibited the ability of *Pseudomonas* PS27 cells to grow and develop as a biofilm (Figure 6) at a concentration (250 μg mL^−1^) that greatly impaired *Pseudomonas* PS27 planktonic growth (Figure 5a). Specifically, 15HQ-POSS/5CuTi completely impaired cell adhesion, thus successfully impairing biofilm formation and proving its suitability as an antifouling agent, whereas 10HQ-POSS/Ti still allowed cell adhesion events to occur, likely due to the slow kinetic release and lower amount of HQ-POSS adsorbed onto the titania support compared to the copper/titania one.

The evidence reported here on the biological activity of short-chain QAS-functionalized POSS structures even when adsorbed on a suitable support sheds an interesting light on the exploration of new highly functional materials from a biocidal perspective.

## 4. Conclusions

Two imidazolium-based POSS compounds were easily prepared by a reaction between octakis(3-chloropropyl)octasilsesquioxane and four equivalents of 1-hexylimidazole or 1-decylimidazole. The use of four equivalents of imidazole compounds was dictated by the fact that an extensive quaternization may feature derived compounds with low antimicrobial potential. HQ-POSS (hexyl-imidazolium quaternized POSS) showed excellent antimicrobial activity toward Gram-positive and Gram-negative microorganisms, compared to DQ-POSS (decyl-imidazolium quaternized POSS). The absence of antimicrobial activity by the latter compound could be ascribed to agglomeration phenomena occurring between the alkyl chains of QASs via ionic and van der Waals interactions. Thus, the best working formulation, HQ-POSS, was adsorbed in different amounts onto the surface of titania and 5 wt% copper/titania for an applicative perspective. Two of these materials (i.e., 10HQ-POSS/Ti and 15HQ-POSS/5CuTi) strongly inhibited the ability of *Pseudomonas* PS27 cells to grow and develop as a biofilm at a concentration of 250 μg mL^−1^. In particular, 15HQ-POSS/5CuTi completely impaired cell adhesion, thus successfully impairing biofilm formation, which represents a major biofouling cause in aquatic environments, and proving its suitability as a potential antifouling agent.

## Data Availability

Not applicable.

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
