# Peer review of "Antifouling Systems Based on a Polyhedral Oligomeric Silsesquioxane-Based Hexyl Imidazolium Salt Adsorbed on Copper Nanoparticles Supported on Titania"

_nanomaterials, 2023, doi:10.3390/nano13071291_

Round 1
Reviewer 1 Report
Line 18 and all subsequent cases in the text 5wt%Cu change to: 5 wt% Cu
Line 49 ca. should be in italics
Line 55 give full name of the acronym PDMS
Line 69 p should be in italics oligo(p-phe-nylenevinylene)
Line 99 leave space Cu(NO3)2.2.5 H2O
Table 1 leave spaces between no. wt% Cu
Line 123 give the full name of the acronym BJH
Line 184 change to ...were prepared by solubilizing ...
Line 266 (DQ-POSS did not negatively affect instead bacterial survival at all tested concentrations) ??? does not make sense, please rephrase
The sentence in lines 321-322 (A decrease in the surface occurs after copper deposition and calcination compared to both the as-received oxide and the calcined one.) ??? does not make sense, please rephrase
The sentence in lines 387-385 is far too long and cannot be followed by the reader. This should be changed to 2 or 3 smaller sentences.
Also, the sentence in lines 402-406 is far too long and cannot be followed by the reader. This should be changed to 2 or 3 smaller sentences.
The conclusions section needs to expand a bit more. Also, the font size is smaller than the previous pages.
Author Response
Dear Reviewer,
we would like to thank you for corrections and useful suggestions.
We have done the corrections listed.
In addition we have modified the text as suggested in your revision. All the changes are visible in the revised version of the manuscript.
The conclusion section was expanded a little bit more.
Best regards
Michelangelo Gruttadauria
Reviewer 2 Report
1. The structure of abstract should be improved. More results data and conclusions should be added in the abstract section.
2. The application of this system and similar results can be also included in the introduction section.
3. The data analysis should be added in the materials and methods section.
4. The data in table 3 and table 4 should be expressed as average and SD.
5. The effect mechanism should be discussed deeply by cited more related articles.
Author Response
Dear Reviewer,
we would like to thank you for corrections and useful suggestions.
We have done the corrections listed.
The abstract has been improved including more results data and conclusions.
In the introduction section similar results based on POSS containing quaternary ammonium salts are already enclosed as well as other quaternary ammonium salts.
Data in Table 3 and 4 have been expressed as average values. In the experimental part we have specified that the precision on the mass loss values was ± 0.001 and ± 2% on the wt% of HQ-POSS in the samples. Concerning textural values (SSA, pore volume and mean pore size) listed in Table 3, it has been reported in the experimental part that such values are quoted with a precision of ± 1%, ± 0.01 on pore volume and ± 0.1 on the mean pore size values, respectively
The mechanism of quaternary ammonium salts has been thoroughly discussed in the manuscript with several references. All changes are visible in the revised version of the manuscript.
Best regards
Michelangelo Gruttadauria
Reviewer 3 Report
This study deals with the development of antifouling coating deposited on copper-modified titania. There are numerous characterizations, and globally the work is well-executed. The paper needs major revisions before publication, here are some comments:
- In the results and discussion, it would be more fluent to have subtitles to divide the text.
- The titania is modified with 5wt% copper but there is actually no characterization showing the presence of copper as ICP and XRD.
- The authors used the pronoun “we” in the text, it must be avoid in a scientific paper.
- In the Table 3, can you uniform the number after the coma for 10HQ-POSS/Ti sample ?
- In Tables 3 and 4, are the 4 numbers after coma significant ?
- It would be nice to have a comparison of your antifouling results with the literature.
Author Response
Dear Reviewer,
we would like to thank you for corrections and useful suggestions.
We have done the corrections listed.
We have added subtitles in the results and discussion section.
- The titania is modified with 5wt% copper but there is actually no characterization showing the presence of copper as ICP and XRD.
Response : according to the reviewer suggestion we have added the missing information in the revised version.
Elemental analysis of the 5CuTi sample was carried out by Atomic Emission Spectroscopy (MP-AES 4200 Agilent technologies), after treatment in acidic solution with H2SO4 and HNO3 at 250 °C. The real chemical composition corresponded well to the nominal one, 5.0 wt% ±5%. Moreover, the XRD analysis of the 5CuTi sample was performed. The data are commented in the discussion (lines 358-362) as follows “Accordingly, XRD pattern registered on the sample 5CuTi revealed crystalline peaks assigned to the CuO phase along with the typical features of titania, as anatase and rutile, which were confirmed by cross-referencing with ICSD files (see supporting information, Figure S8).
In the Table 3, can you uniform the number after the coma for 10HQ-POSS/Ti sample ?
Response: it was done.
- In Tables 3 and 4, are the 4 numbers after coma significant ?
Response: the data in Table 3 and 4 have been expressed as average values. In the experimental part we have specified that the precision on the mass loss values was ± 0.001 and ± 2% on the wt% of HQ-POSS in the samples. Concerning textural values (SSA, pore volume and mean pore size) listed in Table 3, it has been reported in the experimental part that such values are quoted with a precision of ± 1%, ± 0.01 on pore volume and ± 0.1 on the mean pore size values, respectively.
- The authors used the pronoun “we” in the text, it must be avoid in a scientific paper.
Response: we have modified these sentences.
Some comparisons of our antifouling results with the literature are already reported in the manuscripts, however, a direct comparison is very difficult as many parameters are different. Nevertheless, additional references have been added as well as comments on the mechanism of action.
All changes are visible in the revised version of the manuscript.
Best regards
Michelangelo Gruttadauria
Round 2
Reviewer 3 Report
The comments were well addressed and the paper can be accepted for publication.